# The Role of Myocardial Perfusion Imaging in the Prediction of Major Adverse Cardiovascular Events at 1 Year Follow-Up: A Single Center’s Experience

**DOI:** 10.3390/jpm13050871

**Published:** 2023-05-22

**Authors:** Paraskevi Zotou, Aris Bechlioulis, Spyridon Tsiouris, Katerina K. Naka, Xanthi Xourgia, Konstantinos Pappas, Lampros Lakkas, Aidonis Rammos, John Kalef-Ezra, Lampros K. Michalis, Andreas Fotopoulos

**Affiliations:** 1Nuclear Medicine Department, University Hospital of Ioannina, 45500 Ioannina, Greece; 2Second Department of Cardiology, Faculty of Medicine, School of Health Sciences, University of Ioannina and University Hospital of Ioannina, 45500 Ioannina, Greece; md02798@yahoo.gr (A.B.);

**Keywords:** myocardial perfusion imaging, SPECT study, coronary artery disease, mortality, major adverse cardiovascular events

## Abstract

Background: Myocardial perfusion imaging via single-photon emission computed tomography (SPECT MPI) is a well-established method of diagnosing coronary artery disease (CAD). The purpose of this study was to assess the role of SPECT MPI in predicting major cardiovascular events. Methods: The study population was composed of 614 consecutive patients (mean age: 67 years, 55% male) referred for SPECT MPI due to symptoms of stable CAD. The SPECT MPI was performed using a single-day protocol. We conducted a follow-up on all patients at 12 months via a telephone interview. Results: The majority of our patients (78%) presented findings suggestive of reversible ischemia, fixed defects or both. Extensive perfusion defects were found in 18% of the population, while LV dilation was found in 7%. During the 12-month follow-up, 16 deaths, 8 non-fatal MIs and 20 non-fatal strokes were recorded. There was no significant association of SPECT findings with the combined endpoint of all-cause death, non-fatal MI and non-fatal stroke. The presence of extensive perfusion defects was an independent predictor of mortality at 12 months (HR: 2.90, 95% CI: 1.05, 8.06, *p* = 0.041). Conclusions: In a high-risk patient population with suspected stable CAD, only large reversible perfusion defects in SPECT MPI were independently associated with mortality at 1 year. Further trials are needed to validate our findings and refine the role of SPECT MPI findings in the diagnosis and prognosis of cardiovascular patients.

## 1. Introduction

Cardiovascular disease (CVD) is the leading cause of death in the western world, and coronary artery disease (CAD) is the most common form of CVD [1]. Patients with symptoms suggestive of CAD undergo a series of evaluations based on their relevant risk, ultimately assessing the need for invasive coronary angiography and the need for treatment. Diagnostic methods for coronary artery disease provide information both on the presence of angiographically significant CAD and also on the clinical prognosis of patients (i.e., mortality, non-fatal myocardial infarction, angina, etc.) [2].

Myocardial perfusion imaging (MPI) via single-photon emission computed tomography (SPECT) is a well-established nuclear cardiology method used to assess the coronary artery blood supply of the left ventricle. Performed in conjunction with a stress test (physical or pharmacological), SPECT MPI is a valuable imaging modality used to diagnose CAD, allowing for the evaluation of the extent and severity of CAD as assessed via coronary angiography [3]. The diagnostic performance of SPECT MPI has been extensively studied, with considerable heterogeneity in the estimated accuracy among studies [4]. Whether SPECT MPI findings may further improve patient risk stratification and determine the occurrence of major clinical events is still a matter of debate [3,5].

The aim of the current single-center study was to evaluate the prognostic role of SPECT MPI, in terms of predicting mortality and other major cardiovascular events at 1 year, in symptomatic patients assessed for suspected stable CAD in the modern era.

## 2. Patients and Methods

### 2.1. Study Population

The study population consisted of 614 patients referred for SPECT MPI in the Nuclear Medicine Department of a tertiary hospital over a year (1 January to 31 December 2017) due to symptoms (i.e., chest pain (CCS class I or II) or dyspnea (NYHA class I-III)) suggestive of suspected stable CAD. Patients who needed a revascularization procedure according to the MPI study results were retrospectively excluded from our analysis. A revascularization procedure based on the reference SPECT MPI study results may have an important impact on the prognosis of the patients and would complicate the association of SPECT MPI findings with the clinical outcomes. Other exclusion criteria were: a recent myocardial infarction (MI) in the last month before the test or a coronary revascularization procedure (percutaneous coronary intervention or coronary artery bypass graft) in the last two months.

### 2.2. Study Protocol

On the day of the test, every patient was interviewed to enable us to record all the data relevant to their cardiovascular history. This information involved cardiac-related symptoms, risk factors for CAD (smoking habits, hypertension, diabetes mellitus, dyslipidemia, obesity, family history of CAD) and a previous history of any diagnosed CVD and its management. The study protocol and the SPECT MPI procedure were explained in detail to the patients, who gave their written informed consent to participate. The study was compliant with the principles of the Helsinki Declaration and with local legislation, and was approved by the Scientific Council and Ethical Committee of the hospital.

We conducted a follow-up on all patients enrolled in the study at 12 months after the SPECT MPI via a telephone interview. The patients—or their next of kin—were asked in detail about the occurrence of clinical events of importance in the time interval since their enrollment in the study. In case a major cardiovascular event was reported, the findings were confirmed via review of the corresponding medical records. The primary endpoint of the study was the composite of all-cause mortality, nonfatal MI and nonfatal stroke. MI and stroke were defined according to the clinical criteria of hospitalization, biochemical blood tests, electrocardiography and brain imaging.

### 2.3. Myocardial Perfusion Imaging Protocol

The same data acquisition parameters were used in all patients. SPECT MPI was performed according to the guidelines of the European Association of Nuclear Medicine and Molecular Imaging (EANMMI) [6]. The single-day protocol was used, with stress and rest images acquired in that particular order, approximately two hours apart. Patients fasted for at least four hours before the exam and abstained from methyl-xanthine beverages and caffeine for 24 h, in order to be prepared to undergo vasodilative stress with dipyridamole, if needed. Discontinuation of medications containing nitrates, calcium channel antagonists and beta-blockers was also applied for 24 h before the test.

Maximal or symptom-limited treadmill exercise (Bruce protocol) was selected for those patients who were capable of completing this type of stress test. Intravenous pharmacological stress was applied to patients with limited physical tolerance or with contraindications for treadmill exercise. In those patients, the vasodilative substance dipyridamole (0.56 mg/kg body weight) was mostly used, either alone or in combination with a single-stage 3 min Bruce treadmill exercise. In a small proportion of patients who were ineligible to receive dipyridamole, an inotropic pharmacological stress was carried out with the IV infusion of dobutamine. All studies were performed with Technetium Tc 99m tetrofosmin (TF) (Myoview, GE Healthcare AS, Oslo, Norway), an MPI radiopharmaceutical that was labeled in-house according to the manufacturer’s instructions.

### 2.4. Visual Analysis of Myocardial Perfusion

The recorded raw tomographic data under stress and rest were processed and reconstructed in the three planes (short axis, horizontal long axis and vertical long axis) according to established algorithms, and assessed by two experienced board-certified nuclear medicine physicians. The readers were not blinded to the clinical information and reported jointly to reach an agreement according to routine clinical reporting protocols; possible discrepancies were resolved via consensus. A perfusion study was interpreted visually as normal when bearing no or only borderline equivocal findings. The presence of reversible perfusion deficits was characterized as ischemia, while fixed (irreversible) deficits corresponded either to myocardial necrosis (scar) or to a gravely ischemic (hibernating) myocardium. Finally, studies with coexisting ischemia and irreversible deficits were characterized as having mixed deficits. Further evaluation of each SPECT study involved the estimation of the extent of abnormal left ventricular perfusion by the number of involved myocardial segments, graded on a 3-point scale as: absent or negligible (<1 segment or <5% of the left ventricular myocardium), small to medium (1–4 segments or 5–20% left ventricular myocardium) or large (≥5 segments or >20% left ventricular myocardium) [7].

### 2.5. Statistical Analysis

Continuous data are presented as mean values ± standard deviation, while dichotomous data are presented as numbers (percentage). The association of SPECT MPI findings, as well other clinical parameters, with clinical outcomes was assessed using Cox’s Regression analysis, and Hazard Ratio (HR) estimates (95% confidence interval) were calculated. In multivariate analysis, only the predictive role of SPECT features with a significant univariate association with clinical outcomes was assessed. Receiver–Operator Curve analysis was used to assess the prognostic value of perfusion defects for mortality. A two-tailed *p* value < 0.05 was used to determine significant associations. All analyses were performed using the software IBM SPSS Statistics version 21 (IBM, Armonk, NY, USA).

## 3. Results

The mean age of the population was 67 years, and most patients were male (55%). The prevalence of established cardiovascular risk factors is shown in Table 1. A history of CAD was present in 34% of our patients. Patients were referred for a SPECT MPI test due to chest pain (62%) and/or dyspnea (76%). The majority of our patients (78%) presented findings in the SPECT study that were suggestive of reversible ischemia, fixed defects or both. Extensive perfusion defects were found in 18% of the patients, while left ventricular dilation was found in only 7% of the population. Based on the SPECT results and the clinical findings, 307 (50%) patients were referred for invasive coronary angiography. The finding of extensive perfusion defects was associated with the presence of significant CAD in coronary angiography (r = 0.136, *p* = 0.017).

During the 12-month follow-up, 16 deaths (2.6%), 8 non-fatal MIs (1.3%) and 20 non-fatal strokes (3.3%) occurred. Of the patients who died, 50% had undergone a coronary angiogram; none of them had significant coronary artery disease. There was no significant association of SPECT findings with the primary endpoint (combined endpoint of all-cause death, non-fatal MI and non-fatal stroke) (Table 2).

The presence of extensive perfusion defects and transient left ventricular dilation was significantly associated with mortality at the 12-month follow-up (Table 2 and Figure 1). After adjustment for confounders, only the presence of extensive perfusion defects remained an independent predictor of mortality at 12 months (Table 3). In the ROC analysis, the presence of large perfusion defects was associated with 1-year mortality with an AUC of 0.632, *p* = 0.072.

## 4. Discussion

Our study involved a very high-risk cohort of consecutive symptomatic patients evaluated for suspected CAD, with almost one third of them having a history of CAD. The study patients were extensively treated with statins, antithrombotic medications and other established modern therapies for CVD. In this group of patients, we recorded a relatively high incidence of major adverse cardiovascular events and deaths (4.6% and 2.6%, respectively, at 1 year; this appears to be higher compared to some studies [8,9], although similar results have recently been reported in other studies in similar populations [10]. In the current study, the SPECT MPI findings were not found to be associated with the combined endpoint of death/non-fatal MI or stroke during a 12-month follow-up. However, a significant association of specific SPECT MPI findings (i.e., large ischemic defects and left ventricular dilatation) with all-cause mortality at 12 months was shown. After adjusting for confounders, only large ischemic defects were found to have an independent association with all-cause mortality; patients with large reversible deficits in the SPECT MPI had a ca. three times higher risk of death at 1 year compared to patients with smaller reversible deficits or no deficits.

Myocardial scintigraphy is an important tool for diagnosing CAD, and its combination with other clinical features has been suggested to lead to the more accurate stratification of prognoses and the determination of subsequent treatment policies [11]. Various recent studies have reported a significant, and probably independent, prognostic role of an abnormal SPECT MPI test in different populations, i.e., patients with diabetes mellitus, chronic kidney disease or suspected CAD, irrespective of the presence of obstructive CAD following angiography [12,13,14,15,16]. In these populations, the presence of a normal SPECT MPI study was associated with a lower occurrence of cardiovascular events. In the current study, only the presence of extensive ischemia was found to be related to all-cause mortality, but not to the occurrence of the combined endpoint (death, MI and stroke). The discordant associations observed among various studies may be attributed to differences in design and population characteristics, in the definitions of the endpoints of interest, as well as in the follow-up duration among studies.

Furthermore, it should be noted that different findings of the SPECT MPI study may provide important information for different outcomes. Besides the established role of fixed and reversible perfusion deficits in the diagnosis of suspected CAD, other findings, such as transient left ventricular dilatation and the presence of large reversible perfusion defects, may offer important prognostic information in terms of clinical events, and most importantly, mortality. Indeed, it has previously been shown that specific thresholds of the extent of SPECT MPI-detected ischemia may have a significant independent prognostic role, and thus, may lead to an improvement in the risk stratification of patients with suspected CAD and potential refinement of treatment in these patients [10,17,18,19]. The presence of transient left ventricular dilation has been associated with the diagnosis of extensive, severe CAD and poor prognosis [20]. In the absence of significant CAD, it may be a sign of severe dysfunction of the coronary microcirculation [21,22].

## 5. Limitations

The study was performed in a tertiary care hospital and reflects the practice settings of a single center in a specific cohort of patients, so generalizability may be limited. The exact cause of mortality was not always easy to define; thus, all-cause mortality was chosen as a clinical outcome instead of cardiovascular or other forms of cause-specific mortality. Although the SPECT MPI interpretation was performed by experienced readers, it only involved qualitative visual assessment of the left ventricular perfusion that was applied routinely in the day-to-day clinical reporting practice of this medical center.

## 6. Conclusions

In conclusion, in this contemporary cohort of a high-risk patient population (treated with modern treatment modalities as suggested by the guidelines) with symptoms suggestive of CAD, SPECT MPI findings of large reversible defects were independently associated with increased mortality 1 year after the study. Several aspects of the SPECT MPI study should be taken into account, not only for CAD diagnosis, but also for its prognosis in various populations. Further trials are needed to validate our findings and refine the role of SPECT MPI findings in the diagnosis and prognosis of cardiovascular patients.

## Figures and Tables

**Figure 1 jpm-13-00871-f001:**
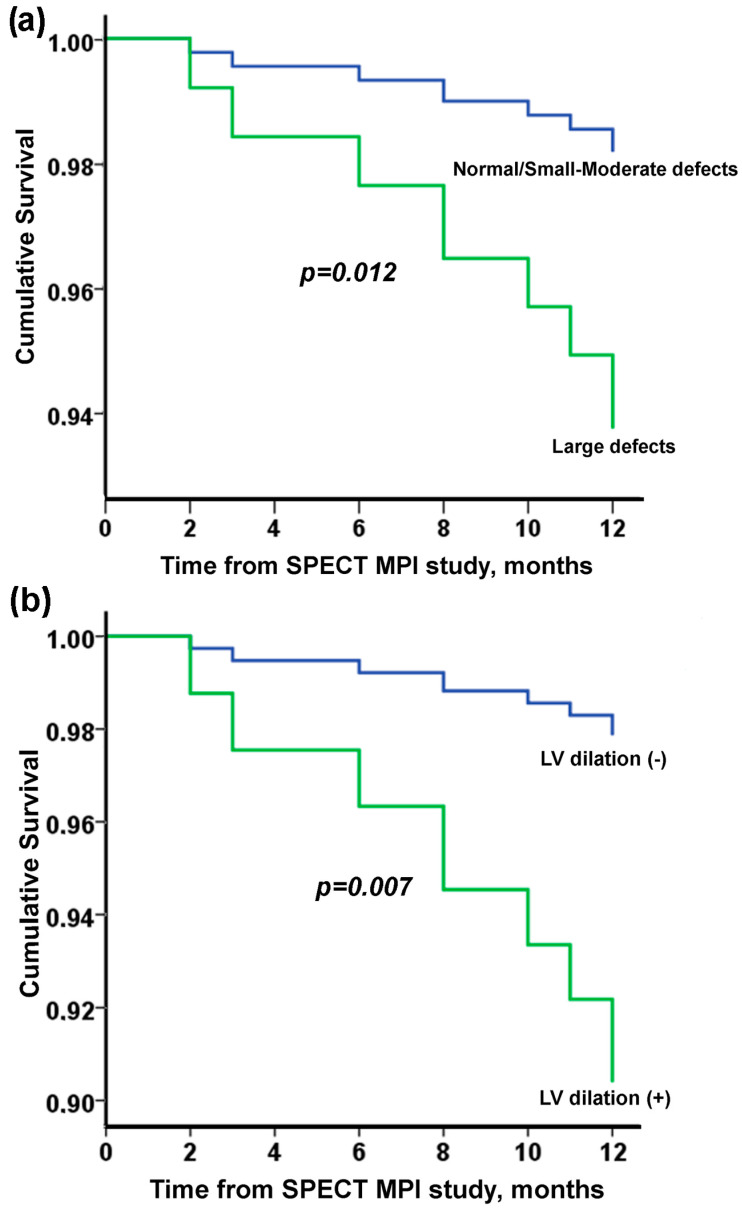
Cumulative survival (Kaplan–Meier curves) according to SPECT MPI study findings: (**a**) extent of defects and (**b**) presence of LV dilation.

**Table 1 jpm-13-00871-t001:** Descriptive characteristics of enrolled patients and results of myocardial perfusion imaging SPECT study.

Age, years	67 ± 10
Male gender, n (%)	340 (55)
Body mass index, kg/m^2^	29.1 ± 4.8
Waist circumference, cm	111 ± 12
Chest Pain (CCS class I-II), n (%)	383 (62)
Typical pain	108 (17)
Atypical pain	215 (35)
Non-angina pain	60 (10)
Dyspnea, n (%)	466 (76)
Low workload—NYHA III	340 (55)
Moderate workload—NYHA II	73 (12)
High workload—NYHA I-II	53 (9)
Currently smoking, n (%)	113 (18)
Hypertension, n (%)	477 (78)
Dyslipidemia, n (%)	469 (76)
Diabetes mellitus, n (%)	205 (33)
History of coronary artery disease, n (%)	208 (34)
History of myocardial infarction	115 (19)
History of Percutaneous coronary interventions	138 (23)
History of Coronary artery bypass surgery	46 (8)
History of atrial fibrillation, n (%)	79 (13)
History of heart failure, n (%)	38 (6)
History of stroke, n (%)	53 (9)
History of peripheral arterial disease, n (%)	75 (12)
Medications, n (%)	
Beta blockers	335 (55)
ACE-I/ATIIR blockers	395 (64)
MRA	33 (5)
Statins	418 (68)
Nitrates	55 (9)
Calcium channel blockers	210 (34)
Diuretics	220 (36)
Antiplatelets	299 (49)
Anticoagulants	72 (12)
Abnormal findings in SPECT MPI study, n (%)	478 (78)
Reversible perfusion defects only	265 (43)
Fixed perfusion defects only	52 (9)
Mixed findings	161 (26)
Extent of perfusion defects in SPECT MPI study, n (%)	
Small/moderate	366 (60)
Large	112 (18)
Left ventricular dilation, n (%)	42 (7)

**Table 2 jpm-13-00871-t002:** Univariate associations of SPECT MPI findings with clinical outcomes at 12-month follow-up.

	HR	95% CI	*p* Value
Combined endpoint			
Normal study	Ref.		
Reversible perfusion defects only	1.25	0.52, 3.01	0.624
Fixed perfusion defects only	1.97	0.62, 6.19	0.249
Mixed findings	1.72	0.69, 4.26	0.242
Normal study/small or moderate defects	Ref.		
Large defects	1.56	0.79, 3.03	0.204
Normal study	Ref.		
Left ventricular dilation	1.41	0.51, 3.96	0.509
Mortality			
Normal study	Ref.		
Reversible perfusion defects only	0.68	0.15, 3.04	0.614
Fixed perfusion defects only	-		-
Mixed findings	2.57	0.70, 9.49	0.157
Normal study/small or moderate defects	Ref.		
Large defects	3.56	1.32, 9.55	0.012
Normal study	Ref.		
Left ventricular dilation	4.74	1.53, 14.71	0.007

**Table 3 jpm-13-00871-t003:** Associations of various studied parameters with all-cause mortality at 12-month follow-up.

	Univariate Analysis			Multivariate Analysis		
Mortality	HR	95% CI	*p* Value	HR	95% CI	*p* Value
Large defects	3.56	1.32, 9.55	0.012	2.90	1.05, 8.06	0.041
LV dilation	4.74	1.53, 14.71	0.007			
Age	1.13	1.05, 1.21	0.001	1.13	1.05, 1.22	0.001
Male gender	5.72	1.30, 25.15	0.021			
BMI	0.79	0.69, 0.90	0.001	0.80	0.68, 0.93	0.004
Waist	0.97	0.93, 1.01	0.130			
Chest pain	0.60	0.23, 1.60	0.310			
Dyspnea	1.30	0.76, 2.22	0.341			
Smoking	1.02	0.29, 3.57	0.978			
Hypertension	0.86	0.28, 2.68	0.799			
Diabetes	1.20	0.44, 3.30	0.725			
Dyslipidemia	1.34	0.38, 4.69	0.651			
History of CAD	3.28	1.19, 9.01	0.022			
Heart failure	3.60	1.03, 12.62	0.046			
History of stroke	1.49	0.34, 6.57	0.596			
Non-fatal MI						
Large defects	2.71	0.65, 11.35	0.173			
LV dilation	-	-	-			
Age	1.06	0.98, 1.16	0.150			
Male gender	1.35	0.32, 5.63	0.684			
BMI	1.08	0.95, 1.22	0.258			
Waist	1.06	1.01, 1.12	0.029	1.06	1.01, 1.12	0.029
Chest pain	0.36	0.09, 1.51	0.163			
Dyspnea	0.95	0.19, 4.70	0.948			
Smoking	0.63	0.08, 5.12	0.666			
Hypertension	0.86	0.17, 4.26	0.852			
Diabetes	1.20	0.29, 5.01	0.805			
Dyslipidemia	2.17	0.27, 17.60	0.470			
History of CAD	1.97	0.49, 7.88	0.338			
Heart failure	2.16	0.27, 17.57	0.471			
History of stroke	1.50	0.19, 12.22	0.703			
Stroke						
Large defects	0.49	0.11, 2.12	0.341			
LV dilation	-	-	-			
Age	1.08	1.02, 1.14	0.010	1.08	1.02, 1.14	0.010
Male gender	0.99	0.41, 2.38	0.975			
BMI	0.98	0.89, 1.08	0.674			
Waist	0.97	0.94, 1.01	0.134			
Chest pain	0.60	0.25, 1.43	0.247			
Dyspnea	2.87	0.67, 12.38	0.157			
Smoking	1.50	0.54, 4.12	0.435			
Hypertension	1.64	0.48, 5.59	0.430			
Diabetes	0.50	0.17, 1.48	0.208			
Dyslipidemia	1.25	0.42, 3.72	0.695			
History of CAD	1.61	0.67, 3.88	0.291			
Heart failure	1.72	0.40, 7.41	0.467			
History of stroke	2.74	0.91, 8.18	0.072			

## Data Availability

Data are available from the authors upon request.

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
