# Peer review of "The Role of Myocardial Perfusion Imaging in the Prediction of Major Adverse Cardiovascular Events at 1 Year Follow-Up: A Single Center’s Experience"

_jpm, 2023, doi:10.3390/jpm13050871_

Round 1

Reviewer 1 Report

This study reported that the value of myocardial perfusion imaging in the prediction of major adverse cardiovascular events at 1 year follow-up: a single center’s experience. A total of 614 consecutive patients were enrolled.  Only large reversible perfusion defects in SPECT MPI were independently associated with mortality at 1 year.  The novelty of the study is low. The results were not surprised. Because the risk stratification value of MPI has been extensively proved by many previous studies. 

Of patients who died, 50% had undergone a coronary angiogram; none of them had significant coronary artery disease. It is not clear.

Some reference is not correct. 

The quality of English language is OK. 

Author Response

We thank the reviewer fo the time invested on the review of our manuscript. Below you may find a point-by-point response to the coment/suggestions.

This study reported that the value of myocardial perfusion imaging in the prediction of major adverse cardiovascular events at 1 year follow-up: a single center’s experience. A total of 614 consecutive patients were enrolled.  Only large reversible perfusion defects in SPECT MPI were independently associated with mortality at 1 year.  The novelty of the study is low. The results were not surprised. Because the risk stratification value of MPI has been extensively proved by many previous studies.

We thank the reviewer for the comments. Indeed the prognostic role of MPI has been extensively proven in many previous studies. Currently we replicate these findings in a modern cohort of well treated patients according to modern management guidelines – this is now stated clearly in the Conclusions section (highlighted in yellow). This is quite important as it further supports the significant role of MPI in the management of these patients in the modern era.

Of patients who died, 50% had undergone a coronary angiogram; none of them had significant coronary artery disease. It is not clear.

Thank you for your comment. Currently, we report that of those who died during the follow-up, only 50% had actually undergone a new coronary angiography procedure based on the results of reference MPI test (according to the judgement of their physicians) but none of those 50% had significant epicardial CAD in the angiography. Thus significant epicardial coronary artery stenosis were not probably the major drive of all-cause mortality in our cohort. BUT despite this finding MPI has an important role in the prognosis (probably independent of epicardial CAD).

Some reference is not correct.

Thank you for the suggestion. A thorough revision of our references was made and correction were made where applicable.

Reviewer 2 Report

The authors describe the role of myocardial perfusion imaging in the prediction of major adverse cardiovascular events at 1-year follow-up in a single center experience.

The topic is interesting, here my comments

1. The current CCS ESC guidelines do not support the choice to perform SPECT imaging in patients with a history of typical chest pain with a high-risk profile for CAD. A significant proportion of patients in the study had typical chest pain and risk factors for CAD. According to ESC guidelines, these patients should be referred to the ICA.

2. Patient profile assessment: Interviewing patients for risk factors of CAD is not reliable such as blood samples for detection of diabetes, dyslipidemia, CKD etc. Specify how the risk factors and patients’ profile were defined (e.g. blood draws, medications taken, etc.).

3. Why did you choose a 12-month telephone follow-up? Telephone follow-up is not a reliable method of detecting important cardiovascular events.

4. The primary endpoint is all-cause mortality does not adequately study the correlation between

SPECT findings suggestive of CAD and actual CAD mortality or major adverse cardiovascular events at one year follow-up, as the study claims to analyze.

5. Why did you not consider other endpoints in the analysis? For example, heart failure, re-hospitalizations, etc. Large perfusion defects, which were founded to be correlated with the primary endpoint of all-cause mortality, can also lead to death from heart failure, MI, or other causes. It could be interesting and of additional value to be known.

6. In the statistical analysis, 50% of patients were excluded because they were referred to the ICA (and PCI).

This raises some concerns about the studio design. Were those high-risk patients? What were their characteristics? If so, why were they included in the study and not referred to the ICA?

7. SPECT readers were not blinded. Explain why this choice was made or needed.

English quality is acceptable

Author Response

We thank the reviewer for the time invested on reading our manuscript. Below you may find a point-by-point answer to the reviewer's comments/suggestions.

The authors describe the role of myocardial perfusion imaging in the prediction of major adverse cardiovascular events at 1-year follow-up in a single center experience.

The topic is interesting, here my comments

  1. The current CCS ESC guidelines do not support the choice to perform SPECT imaging in patients with a history of typical chest pain with a high-risk profile for CAD. A significant proportion of patients in the study had typical chest pain and risk factors for CAD. According to ESC guidelines, these patients should be referred to the ICA.

Thank you for your comment. The typicality of the chest pain was not recorded in our dataset. Furthermore, only the 2/3 of our patients presented with some kind of chest pain while also >70 % presented with dyspnea which is also an atypical presentation of CAD. The selection of SPECT as a modality for myocardial ischemia testing was made by the attending physician of each patient according to international guidelines and local clinical practice. This is now clearly stated in the limitations section (highlighted in yellow)  

  1. Patient profile assessment: Interviewing patients for risk factors of CAD is not reliable such as blood samples for detection of diabetes, dyslipidemia, CKD etc. Specify how the risk factors and patients’ profile were defined (e.g. blood draws, medications taken, etc.).

Thank you for the suggestion. The definition of risk factors was based on the information provided by the patient, the medications of the patients and hospital’s medical records. This is now clearly stated in the Methods section (highlighted in yellow).

  1. Why did you choose a 12-month telephone follow-up? Telephone follow-up is not a reliable method of detecting important cardiovascular events.

Thank you for your comment. Indeed the accuracy of telephone follow-up has several limitations but has the advantage of limited loss of patients to follow-up. Our aim was to report specific parameters at follow-up to avoid data mistakes and always check the information provided with hospital’s medical records if this was applicable.  

  1. The primary endpoint is all-cause mortality does not adequately study the correlation between SPECT findings suggestive of CAD and actual CAD mortality or major adverse cardiovascular events at one year follow-up, as the study claims to analyze.

Thank you for the comment. The aim of the current study was to evaluate the prognostic role of SPECT MPI in terms of predicting mortality and other major cardiovascular events at 1 year in this cohort of patients. We chose to study only all-cause mortality since the bias in data collection at the telephone follow-up would be minimal. Searching for different causes of mortality in a telephone follow-up is quite difficult with high chances of misinterpreting the provided information.

  1. Why did you not consider other endpoints in the analysis? For example, heart failure, re-hospitalizations, etc. Large perfusion defects, which were founded to be correlated with the primary endpoint of all-cause mortality, can also lead to death from heart failure, MI, or other causes. It could be interesting and of additional value to be known.

Thank you for the suggestion. The reviewer has made a fair point regarding the association of SPECT findings with other outcomes such as heart failure events. But as mentioned above we decided to keep the data collection at follow-up as simple as possible to avoid bias in reporting outcomes. Furthermore, only a minority of our cohort (i.e. 6%) reported a history of heart failure at baseline.

  1. In the statistical analysis, 50% of patients were excluded because they were referred to the ICA (and PCI). This raises some concerns about the study design. Were those high-risk patients? What were their characteristics? If so, why were they included in the study and not referred to the ICA?

Thank you for the comment. Our study was more inclusive that described by the reviewer. Indeed, 50% of our patients underwent a coronary angiography but these were not excluded (these patients did not proceed to subsequent PCIs). Patient who had a PCI performed were a priori excluded from the study. We believe that a revascularization procedure due to the reference MPI study results has an important impact on the prognosis of the patients and would complicate the association of MPI findings with the clinical outcomes. This is now clearly stated in the Methods section (highlighted in yellow). On the other hand, enrolled patients with positive SPECT MPI findings were not referred for revascularization according to the judgment of their attending physicians.   

  1. SPECT readers were not blinded. Explain why this choice was made or needed.

Thank you for your comment. Indeed SPECT MPI readers were not blinded to the clinical information of the participants. This is now stated in the Limitiations section (highlighted in yellow). Clearly the SPECT readers did not know anything about the clinical outcomes which is the main focus of the present study.

Reviewer 3 Report

Very interesting study that analyzes the prognostic prognostic role of SPECT MPI in the prediction of major cardiovascular events at 1 year follow-up in patients with stable coronary artery disease.  Manuscript is interesting. All parts of the manuscript are  well written.

My comments are:

  1. Why did authors retrospectively exclude patients who needed a revascularization procedure due to MPI results? 

  2. Patients with large defects or with transitory left ventricular dilatation are high-risk patients and this finding is well known. Since large perfusion defects are associated with the presence of significant coronary artery disease, how many patients are revascularized? Are they all excluded from further analysis? Did revascularization improve their prognosis? Authors may analyze separately prognosis in revascularized patients and those who were not revascularized. 

  3. If patients with large perfusion defects did not undergo coronary angiography and revascularization, what was the reason?  

  4. 50% of patients had no significant stenosis on coronary angiogram. Did they undergo other diagnostic procedures, like MRI? Maybe they have INOCA? What was their prognosis?

Thank you

Author Response

We thank the reviewer for the time invested on the review of our manuscript. Below you may find a point-by-point response to the reviewer's comment/suggestions.

Very interesting study that analyzes the prognostic prognostic role of SPECT MPI in the prediction of major cardiovascular events at 1 year follow-up in patients with stable coronary artery disease.  Manuscript is interesting. All parts of the manuscript are well written.

My comments are:

  1. Why did authors retrospectively exclude patients who needed a revascularization procedure due to MPI results? 

Thank you for your comment. We believe that a revascularization procedure due to the reference MPI study results has an important impact on the prognosis of the patients and would complicate the association of MPI findings with the clinical outcomes. This is now clearly stated in the Methods section (highlighted in yellow)    

  1. Patients with large defects or with transitory left ventricular dilatation are high-risk patients and this finding is well known. Since large perfusion defects are associated with the presence of significant coronary artery disease, how many patients are revascularized? Are they all excluded from further analysis? Did revascularization improve their prognosis? Authors may analyze separately prognosis in revascularized patients and those who were not revascularized. 

Thank you for the comment. As discussed in your previous comment all patients who underwent a revascularization procedure due to the MPI test results were excluded from our study analysis. Currently, we have no data to present for those patients who were revascularized. On the other hand, enrolled patients with LV dilation in MPI and large perfusion defects were not referred for revascularization according to the judgment of their attending physicians.    

  1. If patients with large perfusion defects did not undergo coronary angiography and revascularization, what was the reason?

Thank you for the comment. As discussed in your previous comment enrolled patients with LV dilation in MPI and large perfusion defects were not referred for revascularization according to the judgment of their attending physicians. Unfortunately, there are no further data available in our records of the follow-up that could answer to your question clearly.

  1. 50% of patients had no significant stenosis on coronary angiogram. Did they undergo other diagnostic procedures, like MRI? Maybe they have INOCA? What was their prognosis?

Thank you for the comment. Based on the SPECT results, the clinical findings and the attending physician’s judgement only 50% patients were referred for invasive coronary angiography. In order to make it clear, the reviewer refers to our finding that of patients who died and had undergone a coronary angiogram (i.e. 50% of those who died during the follow-up) none of them had significant coronary artery disease. This finding supports the notion that significant epicardial coronary artery disease was not probably the drive of all-cause mortality in our cohort.

Round 2

Reviewer 2 Report

I believe the work has not improved enough.  

It has many methodological and conceptual weaknesses. The paper to date does not offer particular novelties in the scientific field.

the English form has been improved

Author Response

Thank you for your comments. There have been some minor changes included in the Revised manuscript following the comments of the Academic Editor.